# Pharyngeal Carriage of Beta-Haemolytic *Streptococcus* Species and Seroprevalence of Anti-Streptococcal Antibodies in Children in Bouaké, Côte d’Ivoire

**DOI:** 10.3390/tropicalmed5040177

**Published:** 2020-11-27

**Authors:** Pacôme Monemo, Nadia Demba, Fidèle S. Touré, Adjartou Traoré, Christelle Avi, Micheline A. N’Guessan, Juste O. Tadet, Arthur R. Gobey, Augustin E. Anoh, Abdoulaye Diarrassouba, Marie N. Tuo, Amadou Cissé, Jasmina Saric, Jürg Utzinger, Honoré Tia, Judith Kouassi-N’Djeundo, Sören L. Becker, Chantal Akoua-Koffi

**Affiliations:** 1Laboratoire de Bactériologie-Virologie, Centre Hospitalier Universitaire de Bouaké, Bouaké, Cote d’Ivoire; dembanadia@yahoo.fr (N.D.); sounanantsf@gmail.com (F.S.T.); adja150488@gmail.com (A.T.); michyna2003@yahoo.fr (M.A.N.); jontadet@yahoo.fr (J.O.T.); anohethyl@yahoo.fr (A.E.A.); adiarrassoubaabdoul@yahoo.fr (A.D.); tuononframarie@yahoo.fr (M.N.T.); cissseamadou15@outlook.fr (A.C.); honoretia@yahoo.fr (H.T.); akouamc@yahoo.fr (C.A.-K.); 2Unité de Formation et Recherche des Sciences Médicales, Université Alassane Ouattara, Bouaké, Cote d’Ivoire; arthurgorich@gmail.com (A.R.G.); jtoumodj@yahoo.fr (J.K.-N.); 3Service de Pédiatrie, Centre Hospitalier Universitaire de Bouaké, Bouaké, Cote d’Ivoire; avichristelle611@gmail.com; 4Laboratoire d’Immunologie, Centre Hospitalier Universitaire de Bouaké, Bouaké, Cote d’Ivoire; 5Swiss Tropical and Public Health Institute, CH-4002 Basel, Switzerland; j.saric@swisstph.ch (J.S.); juerg.utzinger@swisstph.ch (J.U.); 6University of Basel, CH-4003 Basel, Switzerland; 7Service d’Oto-Rhino-Laryngologie, Centre Hospitalier Universitaire de Bouaké, Bouaké, Cote d’Ivoire; 8Institute of Medical Microbiology and Hygiene, Saarland University, 66421 Homburg/Saar, Germany

**Keywords:** clinical microbiology, Côte d’Ivoire, diagnosis, epidemiology, *Streptococcus*

## Abstract

The pharynx of the child may serve as a reservoir of pathogenic bacteria, including beta-haemolytic group A streptococci (GAS), which can give rise to upper airway infections and post-streptococcal diseases. The objective of this study was to determine the prevalence of beta-haemolytic *Streptococcus* spp. in pharyngeal samples stemming from children aged 3–14 years in Bouaké, central Côte d’Ivoire. Oropharyngeal throat swabs for microbiological culture and venous blood samples to determine the seroprevalence of antistreptolysin O antibodies (ASO) were obtained from 400 children in March 2017. Identification was carried out using conventional bacteriological methods. Serogrouping was performed with a latex agglutination test, while an immunological agglutination assay was employed for ASO titres. The mean age of participating children was 9 years (standard deviation 2.5 years). In total, we detected 190 bacteria in culture, with 109 beta-haemolytic *Streptococcus* isolates, resulting in an oropharyngeal carriage rate of 27.2%. Group C streptococci accounted for 82.6% of all isolates, whereas GAS were rarely found (4.6%). The ASO seroprevalence was 17.3%. There was no correlation between serology and prevalence of streptococci (*p =* 0.722). In conclusion, there is a high pharyngeal carriage rate of non-GAS strains in children from Bouaké, warranting further investigation.

## 1. Introduction

The oro- and nasopharynx of the child is frequently colonised by a variety of bacterial species that are part of the physiological resident microbiota. However, the pharynx can also act as a reservoir for pathogenic bacteria that can give rise to major invasive infections, such as meningitis, ear infections, pneumonia and septicaemia [1,2]. In children, the genus *Streptococcus* is of particular relevance as it colonises the upper airways within the first months of life and is present in the nasopharynx of many children at the age of 2 years [3]. Streptococci are Gram-positive cocci and are commonly divided into three main groups in clinical practice, based on their haemolysis characteristics. Alpha-haemolytic streptococci include the facultative pathogenic *Streptococcous pneumoniae* and the so-called “viridans streptococci”, whereas beta-haemolytic streptococci comprise *Streptococcus pyogenes* and other species that can cause tonsillitis and other purulent infections of the pharyngeal tract. Non-haemolytic streptococci include a large group of different species with variable pathogenic potential.

In nurseries and schools, where many children gather, transmission of *Streptococcus* spp. from one individual to another can take place via oropharyngeal and nasopharyngeal secretions [4]. Factors that influence whether or not children are being colonised by streptococci include age, living in close contact to siblings and socioeconomic status [5]. Exposure to tobacco smoke was also found to be associated with an increased carriage rate of some pathogenic bacteria in day-care children [6].

An exact taxonomic classification of beta-haemolytic streptococci may be challenging. A widely employed method of serological grouping is based on a common cell wall carbohydrate antigen of streptococci, the so-called polysaccharide C. Indeed, the Lancefield classification uses this characteristic to divide streptococci into different groups with varying pathogenicity. Group A streptococci (GAS) comprise the species *S. pyogenes*, which is of particular public health relevance and may be naturally present in the nose or throat of healthy individuals, but can also lead to invasive bacterial infection and subsequent complications following infection [7]. There is considerable debate pertaining to the potential clinical relevance of pharyngeal carriage of non-GAS streptococci, and data from resource-constrained settings are scarce [7]. Groups C and G beta-haemolytic streptococci predominantly belong to the species *Streptococcus dysgalactiae* subsp. *equisimilis* (SDSE), which may also cause significant infections in humans, whereas *S. dysgalactiae* subsp. *dysgalactiae* (SDSD) affects almost exclusively animals. It is important to note that the group A antigen may also be present in SDS isolates [8]. Beta-haemolytic streptococci carriage rates in children living in low- and middle-income countries (LMICs) are high, ranging from 10% to 50%. Although most of these children are asymptomatic, they might facilitate transmission within the community [9].

Bouaké is the second largest city in Côte d’Ivoire, located in the centre of the country. Only a few studies have been carried out regarding pharyngeal carriage of pathogenic bacteria with the majority focusing on *Staphylococcus aureus* [10]. The objective of this study was to determine and characterise the pharyngeal carriage rate of beta-haemolytic streptococci and the seroprevalence of antistreptococcal antibodies in children aged 3–14 years in Bouaké.

## 2. Materials and Methods

### 2.1. Ethics Statement

The study was approved by the medical and scientific management commission of the University and Teaching Hospital of Bouaké (UTHB) in its role as institutional ethics committee, in agreement with educational authorities (Regional Directorate of National Education). Children (aged 3–14 years) attending nursery and primary schools were examined and sampled after oral informed consent was provided by their parents and/or legal guardians. Notably, verbal consent was deemed appropriate by the institutional ethics committee due to the high level of illiteracy among the students’ parents and the low risks associated with this research. Particular emphasis was put on a detailed explanation of all procedures by the research team and the teachers prior to sampling.

### 2.2. Study Procedures

The study was carried out in March 2017. Four primary schools in Bouaké were randomly selected from a list of schools near UTHB. Using class registers of approximately 4000 children, eligible individuals were selected using systematic random sampling with a sampling interval of 10 individuals. Baseline data regarding age, sex, number of people living in the same household, exposure to tobacco smoke, and vaccination status were collected using a pre-tested questionnaire. One oropharyngeal swab and one peripheral venous blood sample were obtained from each child and transferred to the microbiology laboratory at UTHB. Exclusion criteria were any signs of acute infection or disease, or absence of oral informed consent from parents/guardians.

### 2.3. Microbiological Methods

Sterile swabs (CLASSIQSwab, Copan; Brescia, Italy) were used to obtain the oropharyngeal specimens. Swabs were cultured on standard agar media supplemented with 5% sheep blood and colistin + nalidixic acid (CNA) to suppress growth of Gram-negative bacteria. The culture media were incubated aerobically in a CO_2_-enriched atmosphere for 48 h. Streptococci were identified based on colony morphology, and catalase and coagulase tests were employed for differentiation from staphylococci (Slidex Pastore STAPH-PLUS, BioRad Laboratories; Hercules, CA, USA). Beta-haemolytic streptococci were subjected to serogrouping (Slidex Strepto Kit, BioRad Laboratories). Notably, we considered only potentially pathogenic beta-haemolytic streptococci for the final analysis, e.g., group F streptococci were not included as they are widely seen as physiological oral microbiota. Antistreptolysin O antibodies (ASO) were measured in blood samples using immunological agglutination (ASLO-TEST, Biolabo SAS; Lyon, France). Antibody testing results ≥200 IU/mL were considered positive test results.

### 2.4. Statistical Analysis

Data entry and processing was done in Excel 16.0 (2016 edition, Microsoft). Statistical analysis was conducted using Epi Info version 7.1.5 (Atlanta, GA, USA). We employed a χ^2^ test at a significance threshold of 5% to estimate a potential relationship between the qualitative ASO seroprevalence and the oropharyngeal carriage of beta-haemolytic streptococci.

## 3. Results

### 3.1. Epidemiological Characteristics

Overall, 400 children were enrolled in this study, and about half of them (49.7%) were in the age group of 7–10 years, followed by children aged 11–14 years (30.0%), and preschool-aged children (3–6 years; 20.3%) (Table 1). The mean age of the children was 9.0 years (standard deviation 2.5 years), and the male to female ratio was 0.95. Most of the children stated to live with 6–12 persons in the same household (87.5%), and about half of the children reported smoke exposure at home.

### 3.2. Oropharyngeal Carriage Rate of Beta-Haemolytic Streptococci

In total, 47.5% of all oropharyngeal swabs showed growth of one or more Gram-positive bacterial species, with beta-haemolytic streptococci being most frequently encountered (26.8%), followed by non-haemolytic streptococci (9.8%) and coagulase-negative staphylocococci (8.3%) (Table 2). *Streptococcus pneumoniae* and *Staphylococcus aureus* were only found in 2.5% and 0.3% of all samples, respectively. The prevalence of beta-haemolytic streptococci was highest in the age group 11–14 years (9.2%).

Among the 107 beta-haemolytic streptococci isolated from oropharyngeal swabs, serogrouping revealed that most belonged to group C streptococci (84.1%), whereas group G streptococci (7.5%) and GAS (4.7%) were less prevalent (Figure 1).

### 3.3. Seroprevalence of Anti-Streptococcal Antibodies

ASO antibodies were detected in the sera of 69 children, resulting in a seroprevalence of 17.3%. Twenty-four of these children had beta-haemolytic streptococci detected in their pharyngeal swabs, composed of 19 children with group C streptococci, four children with group B streptococci and one child with serogroup G. No statistically significant correlation was observed between oropharyngeal carriage of beta-haemolytic streptococci and presence of ASO as determined by a titre ≥200 IU/mL in those children (*p =* 0.722).

## 4. Discussion

In this study, more than a quarter of healthy Ivorian children aged 3–14 years were found to be oropharyngeal carriers of beta-haemolytic streptococci. In contrast to studies focussing on the serogroup carriage of alpha-haemolytic *S. pneumoniae* strains before and after vaccination in resource-constrained settings [11,12], few investigations have been carried out to describe the occurrence of beta-haemolytic streptococci in sub-Saharan Africa. The prevalence found here is considerably higher than reported by other researchers from The Gambia (20.0%), Ethiopia (17.8%) and Gabon (13.4%) [7,13,14]. It has been speculated previously that close contact at home and in schools, coupled with lack of hygienic behaviour, may facilitate the transmission of colonising bacteria [15,16]. However, in the current study, no statistically significant correlation between pharyngeal carriage rates and the number of persons sharing the same household was found.

Carriage of GAS might pose a risk factor for the later development of invasive streptococcal infections and post-streptococcal complications such as acute rheumatic fever or glomerulonephritis [7]. Pharyngeal colonisation with Gram-positive bacteria is common, owing to the ability of streptococci and staphylococci to adhere to human pharyngeal and nasal epithelial cells [17,18]. Notably, by far the most (84.1%) of the 107 beta-haemolytic *Streptococcus* bacteria isolated in the current study were group C serogroups, whereas the better-known GAS accounted for only 4.6%. Bélard and colleagues reported a study from Gabon, in which the prevalence of non-GAS exceeded GAS (8.2% vs. 5.8%); however, we do not know of any study that demonstrates an equally marked difference to the one observed in the present work [7]. A study from India reported even an eight-fold higher prevalence of groups C and G streptococci as compared to GAS in schoolchildren [19]. Yet, the exact clinical consequences of such observations remain to be elucidated, as e.g., a Brazilian study reported an unexpected inverse correlation between a pharyngeal colonisation with group G streptococci and clinical sore throat episodes [20].

Serological testing for streptococci using ASO titres is widely used to investigate previous infections due to the beta-haemolytic species *S. pyogenes* [21]. However, interpretation of the test results remains difficult and shows considerable setting-specificity. Typically, one would expect ASO titres to be elevated for up to six weeks following an infection, but detection for prolonged periods occurs relatively frequently. A follow-up examination several weeks after the initial test might help to document changes in the serological response. Here, the seroprevalence of antistreptococcal antibodies in children was 17.3% and there was no link between pharyngeal carriage of streptococci and elevated ASO titres. A similar previous study in India, during which healthy children were sampled, demonstrated found a considerably higher prevalence (44.5%) than in the study reported here [22]. In this context, it is important to note that, in addition to streptolysin O produced by GAS, related bacterial lysins can also evoke an ASO antibody response.

Our study has several limitations that are offered for consideration. First, we conducted the examinations at only one centre in the central part of Côte d’Ivoire. Generalisation of our findings to other parts of the country and elsewhere in West Africa is therefore to be done with caution. Second, due to resource constraints, we obtained only a single swab per child subjected to quality-controlled microbiological examination. Owing to a certain lack of sensitivity of this diagnostic tool, the “true” prevalence might be higher. Third, bacterial species identification was carried out with conventional methods, but did not employ more sophisticated investigations, such as matrix-assisted laser desorption/ionisation time-of-flight (MALDI-TOF) mass spectrometry, which is not available in Bouaké. Doing so would have helped to further elucidate the distinct streptococcal species. Additionally, agar plates were only incubated aerobically, which might have negatively impacted on growth of species that would benefit from an anaerobic atmosphere (e.g., *Streptococcus anginosus* group). Fourth, we only employed one commercially available test for serogrouping of streptococcal isolates and for the determination of ASO titres, which precludes an evaluation of the test’s diagnostic accuracy characteristics. 

## 5. Conclusions

To our knowledge, this study is the first to report on the oropharyngeal carriage of beta-haemolytic streptococci in children in the central part of Côte d’Ivoire. We found that more than every fourth healthy child was colonised with these bacteria, with group C streptococci accounting for the majority of isolates. The presence of antistreptococcal antibodies and the carriage of GAS did not show any correlation. Our findings call for additional studies to elucidate the possible role of oropharyngeal colonisation for the transmission of infections caused by beta-haemolytic streptococci, with a focus on non-GAS serogroups.

## Figures and Tables

**Figure 1 tropicalmed-05-00177-f001:**
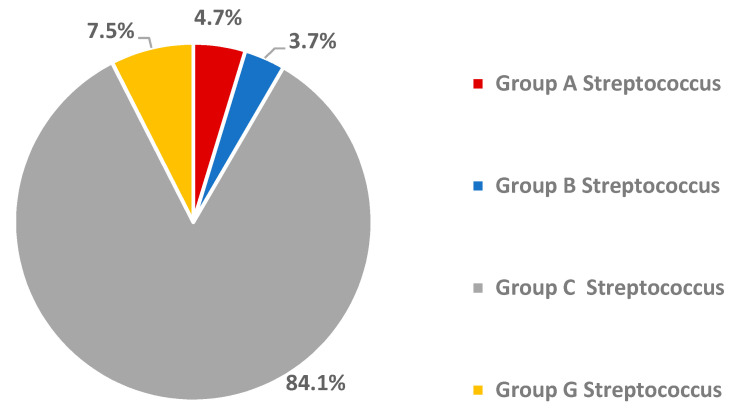
Serogroup distribution among 107 beta-haemolytic streptococci detected in oropharyngeal swabs of healthy children in Bouaké, central Côte d’Ivoire in March 2017.

**Table 1 tropicalmed-05-00177-t001:** Baseline characteristics of 400 children participating in this study pertaining to oropharyngeal carriage of beta-haemolytic streptococci in Bouaké, central Côte d’Ivoire in March 2017.

Characteristic		Number	%
Age (years)	3–6	81	20.2
	7–10	199	49.7
	11–14	120	30.1
Sex	Female	205	51.2
	Male	195	48.8
Schooling level	Kindergarten	28	7.0
	Preparatory class	114	28.5
	Elementary or intermediate class	258	44.5
Number of people living in the same household	1–5 persons	50	12.5
	6–12 persons	350	87.5
Exposure to tobacco	Presence of smokers at home	189	47.3
	Absence of smokers at home	211	52.7
Marital status of parents	Divorced	48	12.0
	Widower	13	3.3
	Living together	339	84.7
Profession	Public sector worker (e.g., teacher, manager)	70	17.5
	Informal sector worker (e.g., traders, unskilled workers)	157	39.2
	Manual worker	125	31.2
	Unemployed	34	8.5
Siblings	1–2	78	19.5
	3–5	256	64.0
	≥6	66	16.5
ASO ^1^ titre	Positive (≥200 IU/mL)	69	17.2
	Negative (<200 IU/mL)	331	82.8

^1^ ASO, anti-streptolysine O antibody.

**Table 2 tropicalmed-05-00177-t002:** Frequency of bacteria identified in oropharyngeal swabs obtained from healthy children aged 3–14 years in Bouaké, central Côte d’Ivoire in March 2017.

Bacteria	Number	%
Beta-haemolytic *Streptococcus* spp.	107	26.8
Non-haemolytic *Streptococcus* spp.	39	9.8
Coagulase-negative *Staphylococcus* spp.	33	8.3
*Streptococcus pneumoniae*	10	2.5
*Staphylococcus aureus*	1	0.3
Negative cultures	210	52.5
**Total**	**400**	**100**

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
