# Peer review of "Pharyngeal Carriage of Beta-Haemolytic Streptococcus Species and Seroprevalence of Anti-Streptococcal Antibodies in Children in Bouaké, Côte d’Ivoire"

_tropicalmed, 2020, doi:10.3390/tropicalmed5040177_

Round 1
Reviewer 1 Report
This is an interesting short communication which would need major improvements before publication in Trop. Med. Infect. Dis. can be considered.
General remarks: Although the correct taxonomic classification of beta-hemolytic streptococci may be less critical in daily clinical diagnostics, it is of great importance for scientific evaluations. Therefore, if only serological grouping based on the cell wall carbohydrate group antigen (not to be confused with the term “typing” as the authors use it) of streptococcal isolated obtained from patient material is applied, the results have to be interpreted with caution. These considerations must be included in the introduction, results, and discussion sections of the manuscript. In particular, it has to be mentioned that
- Groups C and G beta-hemolytic streptococci (the so-called large colony variant) predominantly belong to the species Streptococcus dysgalactiae equisimilis (SDSE) which inhabits the same ecological niches in humans as Streptococcus pyogenes (often just named “group A streptococcus” - GAS), the beta-hemolytic streptococcus with greatest pathogenic potential in man.
- Although the term “Group A Streptococci” (GAS) is largely used in the literature, one should be aware that this expression is based exclusively on the detection of the respective group antigen A in extracts of cultured streptococci. However, the group A antigen may be present in SDSE (which have predominantly the groups C or G antigens – for literature see: Brandt, C. M., Spellerberg, B. (2009) Human infections due to Streptococcus dysgalactiae subspecies equisimilis. Clin Infect Dis 49 (5):766-72. doi:10.1086/605085) or rarely in isolates of the anginosus group.
- It is of importance in which atmosphere (aerobically with or without added CO2 or anaerobically) the cultures for the diagnosis of beta-hemolytic streptococci are grown. This should be added in the Introduction (chapter 2.3. Microbiological Methods). An anaerobic atmosphere will allow better growth of streptococci from the anginosus group in which isolates with group antigens A, C, G, and F can be found.
Additional detailed remarks:
- Change the format of chapter 2.2. to “normal” instead of italics.
- Use “serogrouping” instead of “serotyping” since the latter expression describes typing of other antigens within a streptococcal species!
- Table 1:
- Correct line skip in line “Number of people Iiving in the same household”!
- ASO titre: How is positive or negative defined; give the threshold value (detection limit) of the test! To my knowledge, this is a quantitative test with linearity between 12.5 and 400 IU/mL. It should be discussed, that in addition to streptolysin O produced by pyogenes (“GAS” as diagnosed here) related bacterial lysins can evoke an ASO-response.
- Results incl. Fig. 1: see above concerning nomenclature!
- “group D Streptococcus should not be included in the beta-hemolytic streptococci here, since these are either misdiagnosed (cross-reaction) or they belong to the equinus / bovis group or are Enterococcus species.
- “group F Streptococcus” belongs to the anginosus group; they grow in small colonies and do not have the same pathogenic potential as the other beta-hemolytic streptococci; they do not cause pharyngitis and can be considered as physiological oral flora. In my opinion, they should not be included in the results of the study.
- Lines 141-142: How did the authors calculate the statistical value for the relation between the carriage of beta-hemolytic streptococci and the “presence of ASO”? Is this based on quantitatively determined ASO titers?
- Discussion: lines 161-162: High rates of non-group A beta-hemolytic streptococci were also reported from other low- and middle-income countries – see for example: Bramhachari, P. V. et al. (2010) Disease burden due to Streptococcus dysgalactiae subsp. equisimilis (group G and C streptococcus) is higher than that due to Streptococcus pyogenes among Mumbai school children. J Med Microbiol 59 (Pt 2):220-3. doi:10.1099/jmm.0.015644-0. This should be included in the discussion.
Reviewer 2 Report
Revision of the Manuscript - Pharyngeal Carriage of Beta-haemolytic Streptococcus Species and Seroprevalence of Anti-streptococcal Antibodies in Children in Bouaké, Côte d’Ivoire
Abstract:
Line 37. Do the authors want to say that they obtained 190 positive cultures? If so, it would be better to write in this way.
Introduction
Line 51. “relevance as is colonizes...” Change “is” for “it”
Line 55. As far as I know, genus and species should be written in full when mentioned by the first time (Strep... pyogenes).
Line 59. “via nasopharyngeal secretions...” I would tend to say that is not only nasopharyngeal but also oropharyngeal secretions, especially by child cough.
Line 70. “streptococci carrier rates in children…” Change “carrier” for “carriage”
Materials and Methods
Line 103. Authors should review journal guidelines. Normally, references (city, state and country) about one given company is mentioned only once.
Line 110. Authors should revise references after “Epi Info”. “Druid Hills” is not a city, it is the name of the a region annexed by Atlanta.
Results
Table 1.
Title should be improved. I suggested to change “participating in a study on the oropharyngeal…” for “participating in this study…”
Please pay attention in the column “Number of people living in the same household”. It seems to have an unnecessary space between lines.
“Profession” of the parents? Please state it.
Line 124. “growth of one or more than one…” “than one” is unnecessary. I suggest to remove it.
Antimicrobial susceptibility testing results are missing.
Discussion
Lines 160-162. I would suggest authors to take a look in the global literature, they will find some interesting articles to compare with this study. Eg. DOI: 10.1128/JCM.02095-10. J. Clin. Microbiol. 2011, 49(1):409.
Line 171. “higher prevalence (44.5%)” of what?
Author Response
Please see the attachment uploaded as reply to comments put forth by Reviewer #1.
Round 2
Reviewer 1 Report
A few further corrections are still necessary:
Line 77: Change "SDS" to "SDSE".
Line 132 (Footnote to Table 1): Change "anti-streptolysine" to "anti-streptolysin".
Line 166: Omit "serogroup".
Line 179: Replace "serogroups" by "streptococci".